# One Month of Brief Weekly Magnetic Field Therapy Enhances the Anticancer Potential of Female Human Sera: Randomized Double-Blind Pilot Study

**DOI:** 10.3390/cells14050331

**Published:** 2025-02-23

**Authors:** Jan Nikolas Iversen, Yee Kit Tai, Jasmine Lye Yee Yap, Rafhanah Banu Binte Abdul Razar, Viresh Krishnan Sukumar, Kwan Yu Wu, Melissa Gaik-Ming Ooi, Marek Kukumberg, Sabrina Adam, Abdul Jalil Rufaihah, Alfredo Franco-Obregón

**Affiliations:** 1Department of Surgery, Yong Loo Lin School of Medicine, National University of Singapore, Singapore 119228, Singapore; nikolas.iversen@u.nus.edu (J.N.I.); surylyj@nus.edu.sg (J.L.Y.Y.); rafrazar@nus.edu.sg (R.B.B.A.R.); e0309202@u.nus.edu (V.K.S.); lesleywu@nus.edu.sg (K.Y.W.); surraj@nus.edu.sg (A.J.R.); 2Institute of Health Technology and Innovation (iHealthtech), National University of Singapore, Singapore 117599, Singapore; 3Biolonic Currents Electromagnetic Pulsing Systems Laboratory (BICEPS), National University of Singapore, Singapore 117599, Singapore; 4NUS Centre for Cancer Research, Yong Loo Lin School of Medicine, National University of Singapore, Singapore 117599, Singapore; melissa_ooi@nuhs.edu.sg; 5Department of Medicine, Yong Loo Lin School of Medicine, National University of Singapore, Singapore 119228, Singapore; 6Department of Haematology-Oncology, National University Cancer Institute, National University Hospital, Singapore 119074, Singapore; 7Healthy Longevity Translational Research Programme, Yong Loo Lin School of Medicine, National University of Singapore, Singapore 119228, Singapore; marek.kukumberg@nus.edu.sg (M.K.); med.sabrina@nus.edu.sg (S.A.); 8School of Applied Sciences, Temasek Polytechnic, Singapore 529757, Singapore; 9Department of Physiology, Yong Loo Lin School of Medicine, National University of Singapore, Singapore 117593, Singapore

**Keywords:** soluble biomarkers, peripheral artery disease, metastasis, epithelial–mesenchymal transition, angiogenesis, breast cancer

## Abstract

Preclinical studies have shown that the blood from female mice exposed weekly to magnetic fields inhibited breast cancer growth. This double-blind randomized controlled trial investigated whether analogous magnetic therapy could produce similar anticancer sera from human subjects. Twenty-six healthy adult females (ages 30–45) were assigned to either a magnetic therapy group, receiving twice weekly 1 mT magnetic exposures (10 min/session) for 4 weeks, or a control group, who underwent identical sham exposure. Blood sera were evaluated for their capacity to modulate breast cancer-related cellular responses and epithelial–mesenchymal transition. The sera from the magnetic therapy group subjects exhibited significant anticancer effects that were strongest one month after the last magnetic exposure, whereas the sera from unexposed females or unexposed or exposed males showed no effect. Female sera from the magnetic therapy group (*n* = 12) reduced breast cancer cell proliferation (16.1%), migration (11.8%) and invasion (28.2%) and reduced the levels of key EMT markers relative to the control sera (*n* = 14). Magnetic therapy modulated the serum levels of angiogenic and myogenic biomarkers in a manner consistent with improved cancer management. Muscle-targeted magnetic therapy holds the potential to enhance the anticancer properties of human blood via an adaptive process, akin to exercise training.

## 1. Introduction

Cancer remains a global healthcare threat despite recent advancements in cancer therapeutics and in the understanding of cancer progression. In 2022, the world saw approximately 20 million new cancer cases and 10 million cancer-related deaths [1]. Cancer has overtaken cardiovascular disease as the leading cause of death in first world countries, such as Singapore and the United States, with mortality doubling that of cardiovascular disease [1]. Breast and lung cancers were the most frequently diagnosed amongst women and men, respectively [1]. However, whereas the incidence of lung cancer in men has declined (−2.6%) due to reduced smoking, the incidence of breast cancer continues to rise [2]. Last year, amongst women in the United States, breast cancer accounted for an estimated 31% and 21% of all new cancer cases and cancer deaths, respectively.

Sedentary lifestyles significantly increase the risk of non-communicable diseases (NCDs), including those for breast cancers [1,2]. Notably, high-income countries experience twice as great incidence rates for breast cancer than low-income countries, which is closely associated with the predominance of sedentary work, unhealthy diets and sustained levels of emotional stress and anxiety [3]. Aerobic exercise has been consistently shown to reduce the risks of developing many cancer types, including breast cancer, and is associated with improved overall survival [4]. These benefits of exercise are predominantly attributed to activity-dependent skeletal muscle adaptations to a more oxidative phenotype, which include increased mitochondrial biogenesis, enhanced fat utilization, mitigated inflammation and improved insulin sensitivity [5]. The metabolic adaptations induced by exercise are largely orchestrated by secreted muscle-derived factors [6] that mediate muscle-organ crosstalk [5]. Such exercise-induced exerkines/myokines have been shown to exert anti-tumorigenic effects by inhibiting malignant cell proliferation, invasion, migration and epithelial–mesenchymal transition [7].

Despite these noted benefits of exercise, a significant portion of the population is faced with physical constraints that limit their capacity to exercise. These individuals include those who are frail or whose physical functionality is severely impaired due to cancer progression or the adverse side effects associated with cancer treatment such as chemotherapy, or other debilitating health conditions. This unmet need was hence the existence of non-invasive and drug-free methodologies that can effectively recapitulate the metabolic benefits of exercise, but with minimal physical exertion and time commitment.

Recent studies have shown that a particular form of magnetic stimulation may represent a gentle manner of stimulating muscle mitochondrial respiration that can reproduce a subset of the anti-inflammatory benefits typically ascribed to exercise. Brief (10 min) exposure to low-energy pulsed electromagnetic fields (PEMFs) (1 mT) has been shown to enhance myogenesis and mitochondrial respiration in vitro [8] and in vivo [9] as well as in preliminary human trials [10]. These low-energy magnetic fields activate the PGC-1α transcriptional pathway governing mitochondriogenesis, which is crucial for oxidative muscle expression through an adaptive mitohormetic response [5]. An identified response limb of the PGC-1α transcriptional pathway is the mobilization of the muscular secretome that is strongly activated by this magnetic paradigm [11]. Conditioned media harvested from differentiated muscle cells after a single magnetic exposure, or serum collected from mice exposed weekly, inhibited breast cancer cell growth, migration and invasion [6]. Notably, serum collected from mice after 8 weeks of once-weekly magnetic exposure (10 min per session for a total of 80 min) exhibited a greater capacity to inhibit breast cancer cell invasion and migration than serum obtained from mice that had undertaken 20 min of aerobic exercise twice weekly (320 min of total exercise) [6]. As the evaluated mouse blood was collected one week after the last magnetic exposure, the reported anticancer effects alluded to an adaptive response of muscle to the treatment, like exercise training. These studies suggested that an analogous form of magnetic therapy may potentially offer a novel exercise-mimetic approach to assist in human breast cancer management.

This double-blinded, randomized pilot trial aimed to investigate the potential for muscle-targeted magnetic therapy to enhance the anticancer properties of blood in healthy individuals. Participants (30 to 45 years of age) were randomly assigned to either a PEMF therapy group, receiving 1 mT PEMF sessions to alternating thighs twice weekly for 4 weeks (80 min of total exposure), or a sham exposure group. Blood sera collected from both groups were compared for their capacity to suppress in vitro malignancy and changes in soluble biomarker levels using multiplex assays.

## 2. Materials and Methods

### 2.1. Trial Design and Randomization

The double-blind, randomized pilot trial for healthy volunteers was conducted in collaboration with the National University Hospital (NUH) between May and December 2024. The study protocol was approved by the NHG (National Healthcare Group, Singapore) Domain Specific Review Board (DSRB) (NHG DSRB Ref: 2022/00928) on 26 January 2024. This pilot study is part of an oncology trial investigating the effect of magnetic therapy on hematological patients undergoing prolonged hospitalization (NCT06744764). All procedures adhered to the Declaration of Helsinki and written informed consent was obtained from all participants. A total of 39 healthy human subjects were recruited by the Investigational Medicine Unit (NUH) based on predefined inclusion and exclusion criteria. Inclusion criteria included participants meeting physical activity guidelines as well as being between the ages of 30 and 45 years. Exclusion criteria included pacemakers, pregnancy, psychological or chronic illnesses and nutritional inadequacy.

While in operation, the PEMF device (QuantumTx, Singapore) does not produce any heat or noticeable sensation, allowing for the blinding of participants. All participants were managed identically throughout the trial and placed their limbs into the device twice per week for 10 min with the only exception that the control cohort did not receive device-generated magnetic fields (sham exposure). Participants were randomly assigned to either the PEMF (20) or control (19) cohorts by block randomization using a web-based blinded code list as previously described [12]. The codes were sequentially embedded into each personalized RFID card to deliver either the active or sham treatments without the knowledge of the subject, study coordinator or outcome assessors of treatment group allocation. For the sham exposure condition, the allocated RFID card also activated the device without creating magnetic fields. Therefore, both active exposure and sham scenarios produced similar visual and audio backgrounds. Moreover, using an online sound meter (Youlean.co, Belgrade, Serbia; free mobile copy), both control (sham) and PEMF treatments were found to produce an average sound level of 56 decibels during operation. These measures ensured interventional double blinding. Unblinding was only carried out after the analysis of the blood samples was completed. Four participants withdrew from the trial. Rerecruitment was conducted to maintain an adequate sample size resulting in a final cohort of 17 participants in the PEMF therapy group and 19 in the control group.

### 2.2. Magnetic Intervention

PEMF therapy was administered using the leg coil as previously described [10,11,12]. Succinctly, the PEMF signal consisted of a barrage of magnetic pulses of 6 ms duration applied at a repetition rate of 50 Hz and a flux density of 1 mT (milliTesla). Each 6 ms burst consisted of a series of 20 consecutive asymmetric pulses of 150 µs on and off duration with an approximate rise time of 17 T/s. Participants received 10 min of 1 mT exposure twice weekly to alternating thigh muscles for a total of 4 weeks. Sham control subjects underwent identical procedures to ensure patient blinding, but no magnetic fields were administered. Blood serum and plasma were taken at baseline (week 1), 1 week after the last magnetic exposure (week 5) and after a 4-week washout period (week 8).

### 2.3. Cell Culture and Reagents

Cell lines MCF-7 (HTB-22) and C2C12 were obtained from ATCC (LGC Standards, Teddington, UK), while MCF10A and MDA-MB-231 cells were kindly provided by Dr. Andrew Tan (Nanyang Technological University, NTU) and Dr. Glenn Kunnath Bonney (NUS), respectively. Cells were cultured in the following media: MCF-7: RPMI with 10% FBS (both Gibco, Thermo Fisher Scientific, Waltham, MA, USA); MDA-MB-231 and C2C12: DMEM (HyClone; Danaher, Washington, DC, USA) with 10% FBS. MCF10A: DMEM/F12 with 5% horse serum (HyClone; Danaher, Washington, DC, USA), 20 ng/mL EGF (Peprotech, Thermo Fisher Scientific, Waltham, MA, USA); 0.5 mg/mL Hydrocortisone, 100 ng/mL cholera toxin and 10 ug/mL insulin (Sigma-Aldrich, St. Louis, MO, USA). Cells were passaged every 48 h to maintain <40% confluence and cultured without antibiotics.

### 2.4. DNA Content Analysis and MTT Cell Viability Assay

Cell viability and DNA content were assessed using MTT Cell Proliferation Kit (Roche, Basel, Switzerland) and Cyquant cell proliferation assay (Invitrogen, Thermo Fisher Scientific, Waltham, MA, USA), respectively, as per manufacturer’s instructions. Briefly, cells (1000/well) were seeded in 96-well plates with 8 technical replicates. After 24 h, the media was changed to 5% human serum-supplemented media. Cell viability and DNA content were measured at 24 h or 48 h using a Cytation 5 microplate reader (BioTek, Winooski, VT, USA).

### 2.5. Migration and Invasion Assays

Both assays were performed as previously described [6]. MCF-7 (30,000) cells were seeded into each quadrant of a 4-well culture dish insert (ibidi GmbH, Gräfelfing, Germany). After 24 h, fresh media with 5% human serum was added and gap closure was monitored using light microscopy every 24 h. Average gap distances were analyzed using ImageJ (Version 1.54, U.S. National Institutes of Health, Bethesda, MD, USA). Invasion assay was performed using MDA-MB-231 (195,000) cells seeded on CytoSelect 24-well Cell Invasion Assay Kit (Cell Biolabs, Inc., San Diego, CA, USA) according to the manufacturer’s instructions. The lower chamber was supplemented with 5% human serum and incubated for 48 h in a standard tissue culture incubator. Invaded cells were stained, extracted and analyzed at OD560 using a Cytation 5 microplate reader (BioTek, Winooski, VT, USA).

### 2.6. Western Analysis

Cell lysates were prepared in ice-cold radio immunoprecipitation assay (RIPA) buffer supplemented with protease and phosphatase inhibitors (Sigma-Aldrich, St. Louis, MO, USA) as previously described [6]. A total of 25–50 µg of protein was resolved using 10–12% denaturing polyacrylamide gel electrophoresis and transferred to PVDF membrane (Bio-Rad, Hercules, CA, USA). The primary antibodies were diluted in SuperBlock (TBS; Thermo Fisher Scientific, Waltham, MA, USA) according to Table 1.

Membranes were incubated with anti-rabbit or anti-mouse HRP secondary antibodies (1:3000) for 1 h at room temperature. Chemiluminescence was detected using SuperSignal West Pico or Femto (Thermo Fisher Scientific, Waltham, MA, USA) substrates and analyzed with LI-COR Image Studio (LI-COR Odyssey FC; Li-COR, Lincoln, NE, USA). Protein bands were normalized to GAPDH, α-tubulin or β-actin.

### 2.7. Multiplex Analysis

Sera were analyzed using human-specific myokine and angiogenesis/growth factor magnetic bead panels (Milliplex Map Kit) to quantify the plasma levels of selected analytes (Table 2).

### 2.8. Study Design and Statistical Analyses

This study used control or PEMF sera in in vitro breast cancer assays, focusing on cancer cell viability. Assuming an effect size of 0.67, we determined that 16 samples per group would provide 80% power to detect a 10% difference in cell viability (SD = 15%, α = 0.05). Statistical analysis was performed using GraphPad Prism (v10.2.0), employing one-way and two-way ANOVA with Tukey’s post-testing for group comparisons.

## 3. Results

Thirty-six healthy participants enrolled in the study between 2 May 2024 and 9 Dec 2024, and were randomly allocated into either the control (*n* = 19) or PEMF treatment (*n* = 17) groups (Figure 1). The median age, weight, height, BMI, and physical activity level were identical between the two groups (Table 3).

### 3.1. Magnetic Therapy Conditions Blood Serum with Anticancer Properties

Healthy subjects were administered PEMF sessions (1 mT) twice a week for 4 weeks or sham exposures (0 mT) at identical intervals. Each session (0 mT or 1 mT) lasted for 10 min and was applied to alternating legs as shown (Figure 1C). Blood was collected at baseline (week 1), one (week 5) and four (week 8) weeks after the last treatment session. PEMF sera and control sera were compared for their ability to regulate breast cancer cell viability and behavior using the MCF-7 human breast cancer cell line. The MCF-7 cell line is an estrogen receptor-positive (ER+), progesterone receptor-positive (PR+) and HER2-negative (HER2-) breast cancer cell model representative of a luminal subtype of breast cancer that exhibits the ability to proliferate and form monolayers [13]. Given the estrogen-dependent nature of breast cancer, the viability of MCF-7 cells in culture media supplemented with 5% human sera was initially independently evaluated for sera obtained from male and female subjects to elucidate potential gender-specific effects. Female sera from the magnetically treated cohort exhibited the greatest anticancer activity, reducing cancer viability by ~−4% and ~−14% in sera from weeks 5 and 8, respectively, relative to baseline (Figure 2(Ai)). Relative to control female sera, female PEMF sera obtained at week 8 yielded the largest reduction in cell viability by ~−16%. By contrast, neither PEMF nor control male sera showed any anticancer effects at any of the time points (Figure 2(Bi)). Gender-specific effects of the sera over breast cancer were hence suggested.

An MTT assay was next performed to assess for changes in cellular metabolic efficiency after the administration of the different sera. Female PEMF sera at the 8-week collection timepoint induced the greatest reduction in cancer viability (Figure 2(Aii)). Specifically, week 5 and week 8 female PEMF sera reduced cancer viability by ~−12% and ~−19%, relative to their time-matched female control sera. Consistent with DNA content measurements, male PEMF sera did not impact breast cancer cell metabolic activity (Figure 2(Bii)). Given the greater anticancer potency of the PEMF-conditioned female sera, subsequent analyses were conducted solely using female sera.

Cancer cell invasion was evaluated by monitoring the ability of MDA-MB-231 human breast cancer cells to permeate and invade through a basal membrane over a 24 h period. MDA-MB-231 is a highly aggressive and invasive triple-negative breast cancer cell line and a commonly employed metastatic cell model [14]. MDA-MB-231 cells were cultured in media supplemented with 5% human sera. The provision of week 5 or week 8 PEMF sera reduced the invasiveness of MDA-MB-231 cancer cells, as evidenced by a decrease in the quantity of invading, blue-stained cells (Figure 2(Ci)). Week 8 PEMF sera resulted in the greatest reduction in invasion (−28.2%) relative to week 8 control sera (Figure 2(Cii)). By contrast, control sera did not alter the invasive capacity of cancer cells.

The migratory ability of MCF-7 human breast cancer cells was assessed using a transwell migration assay. The rate of gap closure for MCF-7 cells provisioned with 5% control sera from weeks 1, 5 or 8 was consistent across all timepoints analyzed (Figure 2(Di)). In contrast, MCF-7 cells given week 8 PEMF sera exhibited a significant delay in gap closure, observed across days 2, 3, and 4 by −11.3%, −11.8% and −6.2%, respectively, compared to their respective control sera (Figure 2(Di,ii)). Week 5 PEMF sera showed a slowing in gap closure that did not achieve statistical significance.

To ascertain the level of specificity for cancer, the effects of the PEMF sera and control sera were assessed on two non-cancerous cell lines, MCF10A and C2C12. MCF10A cells are non-malignant human epithelial breast cells and C2C12 cells are murine skeletal myoblasts [6]. The viability of these cells was determined using DNA content analysis in response to either serum. The provision of week 5 or week 8 PEMF sera to MCF10A cells did not alter their viability, being comparable to that observed with the control sera (Figure 2(Ei)). Similarly, the provision of the PEMF sera to C2C12 myoblasts did not affect their viability (Figure 2(Eii)). The female PEMF sera hence appears to preferentially possess anti-breast cancer activity.

### 3.2. PEMF Conditioned-Human Serum Alters Cancer Signaling

The EMT (epithelial-mesenchymal transition) is associated with increased cancer cell malignancy, characterized by enhanced migratory and invasive capacities [15]. The impact of human sera on TGF-β (transforming growth factor beta)-associated EMT markers was evaluated using Western blot analysis in MCF-7 breast cancer cells. MCF-7 cells were exposed to media supplemented with 5% human sera for over 4 days. The week 8 PEMF sera resulted in the significant downregulation of (A) TGFβR2, (B) TWIST, (C) SNAI1, (D) SNAI2 (Slug), (E) β-catenin and (F) Vimentin protein expressions, when compared to week 8 control sera (Figure 3). Week 5 PEMF sera primarily reduced the phosphorylation of SMAD 2/3 as well as the expression of TWIST protein expression.

### 3.3. Serum Changes in Response to PEMF Therapy

Blood plasma collected at baseline (week 1), week 5 and week 8 were analyzed using multiplex panels (Table 2). Appendix A presents the mean concentrations of plasma analytes obtained through multiplex analyses for both groups. After normalization to baseline values (week 1), significant differences were observed between groups and over time. Week 8 PEMF-plasma showed significant reductions in angiogenic biomarkers, including Angiopoietin-2, BMP-9, Endoglin, PLGF, VEGF-A, and VEGF-D, with leptin being the sole exception, increasing in week 8 (Figure 4). The myokine panel revealed significant intergroup differences, with Apelin and LIF higher in PEMF-plasma at week 5, and Fractalkine and FABP3 reduced in week 8 PEMF-plasma compared to control-plasma (Figure 5). No significant changes were observed in the control plasma over time.

## 4. Discussion

This study builds on a prior preclinical study demonstrating an anticancer potential of conditioned media and blood sera collected from PEMF-stimulated muscle cells and mice, respectively. The present study investigated the serological consequences of muscle-targeted PEMF therapy in healthy human subjects. Blood was collected prior to, one week, or four weeks after the final PEMF exposure and was tested for its ability to alter breast cancer properties in vitro. Four weeks of twice-weekly PEMF exposure (10 min per session) of women was sufficient to produce blood sera that was capable of undermining breast cancer cell viability. Interestingly, sera collected one month after the last PEMF exposure (8 weeks) resulted in the greatest reductions in cancer cell growth, migratory and invasive capacities, and TGF-β-associated EMT markers. By contrast, the PEMF sera did not adversely alter the growth of non-malignant cells such as MCF10A (breast epithelial) and C2C12 (myogenic). These results highlight the therapeutic value of PEMF therapy to assist in the clinical management of breast cancer without adversely affecting healthy tissues.

The enhanced anticancer potency of sera collected one month after the last PEMF exposure suggests that it arose from muscle adaptation. Ten-minute exposures to 1 mT PEMFs were previously shown to promote in vitro [8] and in vivo [9] myogeneses toward an oxidative phenotype. An analogous phenotypic shift of muscle is produced by exercise training and is accompanied by an adaptive change of the muscle secretome towards a more anti-inflammatory status [5]. As an in vitro parallel, the priming of differentiating muscle cells with conditioned media obtained from PEMF-exposed myotubes later produced an augmented release of anticancer secretome when the resulting myotubes were stimulated by PEMF exposure [6]. Importantly, unexposed myotubes that had been only primed with magnetic-conditioned media exhibited greater resting release of anticancer secretome, despite never having experienced magnetic exposure directly. The augmented constitutive release of anticancer secretome produced by this in vitro PEMF-preconditioning paradigm alludes to the basis for the sustained anticancer potency of blood one month after receiving the last PEMF exposure and reinforces the relevance of cumulative PEMF stimulation in driving adaptive systemic anticancer responses. Furthermore, the preferential anticancer effects observed for the female sera on breast cancer cells revealed gender-specific differences in the serum response to PEMF therapy that were consistent with previous findings in obtained in female mice following similar magnetic treatment [6].

TGF-β signaling stimulates the expression of the EMT transcriptional factors, Snail/Slug and Twist, that confer stem cell-like properties to tumor-initiating epithelial cells during the progression of breast cancer [15]. Snail/Slug and Twist were also reduced in breast cancer cells by the PEMF sera collected one month after the last PEMF exposure (Figure 3B–D). PEMF-induced TGF suppression will have valuable clinical ramifications for drug-resistant cancers exhibiting stem-like cell characteristics and may help improve chemotherapeutic outcomes in patients with triple-negative breast cancer [16].

### 4.1. Modulation of Systemic Angiogenic and Myogenic Factors

Initial blood analyses revealed that weekly muscle PEMF therapy modulated the serum levels of angiogenic and myogenic factors. Angiogenic factors have been shown to foster both cancer cell proliferation and tumor growth [17]. Angiopoietin-2 (Ang-2), bone morphogenetic protein 9 (BMP-9), endoglin, placental growth factor (PLGF), and vascular endothelial growth factor A and D (VEGF-A and VEGF-D) are all previously described angiogenic regulators [18] that were downregulated by PEMF therapy (Figure 4). The downregulation of these key angiogenic factors in the blood of females in the PEMF therapy group one month after the last magnetic exposure indicates a systemic shift towards a more cancer-restrictive blood composition. These findings align with the anticancer properties observed in the blood of exercise-trained individuals [6,19].

It was previously shown that conditioned media from PEMF-exposed myotubes reversed vascularization and overall size of breast cancer microtumors engrafted onto the chorioallantoic membrane of chicken eggs [6]. The present study expands on these findings by demonstrating that PEMF therapy downregulated the serum levels of known angiogenic regulators. VEGF has been shown to directly promote cancer cell proliferation [20]. The VEGF proteins in combination with angiopoietin-2 also synergistically drive tumor vascularization and growth. The downregulation of these VEGFs following the cessation of PEMF therapy may suppress angiopoietin-2-induced vascular sprouting and tumor growth as previously reported [21]. PEMF therapy also downregulated BMP-9, which plays a complicated role in cancer progression. BMP-9 exhibits dual effects, promoting tumor quiescence and normalizing angiogenesis in breast cancer [22], while also supporting EMT and cancer progression [23]. Elevated serum endoglin levels are associated with metastasis in metastatic breast cancer patients [24]. Endoglin depletion has been shown to reduce cancer-promoting pathways, including TGF-β/SMAD3/VEGF and MAPK/p38 signaling, induced by breast cancer-causing phthalates [25]. Accordingly, endoglin reduction in response to PEMF therapy should produce synonymous anticancer effects. Finally, with reference to angiogenesis, previous studies have shown that PLGF levels correlate with VEGF-A levels in breast cancer tissues, both being elevated in malignant tissues relative to non-malignant tissues. High levels of these factors are linked to shorter recurrence-free survival [26]. Provocatively, metformin, a common diabetes medication, mimics certain biochemical effects of exercise [27], inhibits PLGF expression, and reduces tumor growth and macrophage repolarization [28]. Excessive leptin levels have a dual role, potentially promoting angiogenesis and tumor progression [29] or enhancing immune function and improving tumor burden resolution [30,31].

With reference to myokines (Figure 5), apelin governs skeletal muscle homeostasis and is increased by exercise as well as correlates with physical performance [32]. Loss of apelin signaling can lead to premature cardiac aging and age-related sarcopenia [33]. This study found that plasma apelin levels increased one week after the cessation of PEMF therapy, potentially contributing to a more cancer-restrictive phenotype. Similarly, LIF (leukemia inhibitory factor) was upregulated one week after the final PEMF treatment. LIF has been shown to be an important myokine [34] that promotes muscle regeneration [35]. Electrical stimulation of human myotubes in culture or exercise-induced skeletal muscle contraction upregulated LIF expression that was subsequently secreted into the muscle interstitium, fostering muscle adaptation [34,36]. LIF has been attributed with paradoxical roles in cancer progression, either inhibiting blood-borne leukemic cell proliferation or promoting the development of some solid tumors [37]. Fatty acid-binding protein 3 (FABP3) is involved in lipid transport and tumorigenesis. Elevated levels of FABP3 are indicative of a poor prognosis in several cancers [38] and are increased in peripheral artery disease [39], referring back to the accepted association of angiogenesis in tumor progression [17]. Here, PEMF therapy reduced serum FABP3 levels one month after the cessation of PEMF treatment consistent with a regulatory role of PEMF therapy in lipid metabolism [10] and in agreement with the previously noted anti-angiogenic properties of the female PEMF sera. Fractalkine plays a crucial role in the progression and metastasis of breast cancer, particularly in skeletal dissemination [40]. As a chemokine, fractalkine predominantly interacts with immune cells within the tumor microenvironment to modulate tumor progression [41]. Notably, this study found that adapted muscles following PEMF therapy exhibited reduced circulating levels of fractalkine, which could potentially mitigate systemic inflammatory status and promote an anticancer system milieu. Collectively, the levels of blood-borne pro-angiogenic factors produced by muscles are attenuated by brief PEMF therapy that would potentially beneficially impact breast cancer cell responses. The effect of each individual modulated serum biomarker is hence context-dependent yet alludes to the potential for PEMF therapy to serve as a non-invasive and easy-to-implement approach to target tumor vascularization while supporting systemic health.

### 4.2. Study Limitations and Future Perspectives

This study revealed a potential for PEMF therapy to serve as an adjuvant breast cancer treatment. Nonetheless, the testing of human sera on human breast cancer cell lines is a potential shortcoming of this pilot study. While the MCF-7 and MDA-MB-231 breast cancer cell lines have the experimental advantage of being well-characterized and phenotypically consistent models for breast cancer [13,14], they do not fully recapitulate the complexity of the in vivo tumor microenvironment; the inherent genetic heterogeneity and diversity of primary and secondary tumors will not be recapitulated by these cell lines. Furthermore, the sera evaluated in this study were obtained from healthy participants, which may exhibit modest changes in anticancer factors following PEMF therapy due to existing disease regulation in a healthy body. Consequently, the full therapeutic potential of PEMF therapy may not be adequately realized in the present cohort. Sera from healthy individuals also do not reflect the unique immunological, biochemical and physiological alterations characteristic of cancer patients, including elevated concentrations of chemokines, cytokines and other soluble factors that promote tumor progression [42]. Hence, future studies investigating the serological consequences of PEMF therapy in cancer patients of distinct etiologies are warranted to understand its true therapeutic potential.

Concerning dosage, the present study employed twice weekly PEMF exposure over a course of four weeks for a total exposure time of 80 min for comparative purposes with previous studies [6]. PEMF exposure interval and treatment duration remain to be optimized for best clinical exploitation. Although the anticancer cellular responses produced by the PEMF sera were robust and statistically significant (Figure 2 and Figure 3), serological differences were difficult to detect for some biomarkers. Given the large variability observed in baseline (pre-intervention) values for many of the selected biomarkers (Appendix A), larger cohorts would have been required to achieve statistical significance. Estrogen signaling is a major determinant of breast cancer severity [43]. In this respect, the here-noted gender difference observed in the anticancer potency of the PEMF-conditioned sera merits more detailed investigation. It was previously noted that the composition of the muscle secretome is influenced by gender-specific differences in gene expression, including that responsible for sex hormones [44]. The contribution of the hormonal milieu to the anticancer potency of human blood can be directly tested by artificially adjusting the hormonal composition of the male sera to that of the female status and ascertaining any potential changes in anticancer efficacy with reference to breast cancer. Therefore, how gender, hormonal milieu and cancer types influence the anticancer responses induced by PEMF therapy merit closer examination. Lastly, prudence must be advised when designing magnetic exposure paradigms due to their mitohormetic nature; overexposure should be avoided [5]. More is not necessarily better and has to be balanced against the pursuit of enhanced efficacy. Addressing these remaining issues in future studies will provide a better understanding of the translational potential of PEMF therapy as an adjunctive treatment for general cancer management.

## 5. Conclusions

In conclusion, this study demonstrated that the blood from healthy female subjects having undergone one month of biweekly muscle-targeted PEMF therapy presented with anticancer attributes when tested in vitro. Notably, the anticancer potency of the magnetically conditioned sera was sustained and generally most potent at one month following the last PEMF exposure, suggesting muscle adaptation to the magnetic intervention. Preliminary proteome analyses of the blood sera revealed changes in soluble blood-borne biomarkers implicated in angiogenic and myogenic pathways that were overall consistent with the observed anticancer activity. More comprehensive proteomic and metabolomic analyses will be required to uncover other potentially more relevant anticancer factors as well as to more thoroughly understand the effects of PEMF therapy on humans in health and disease.

## Figures and Tables

**Figure 1 cells-14-00331-f001:**
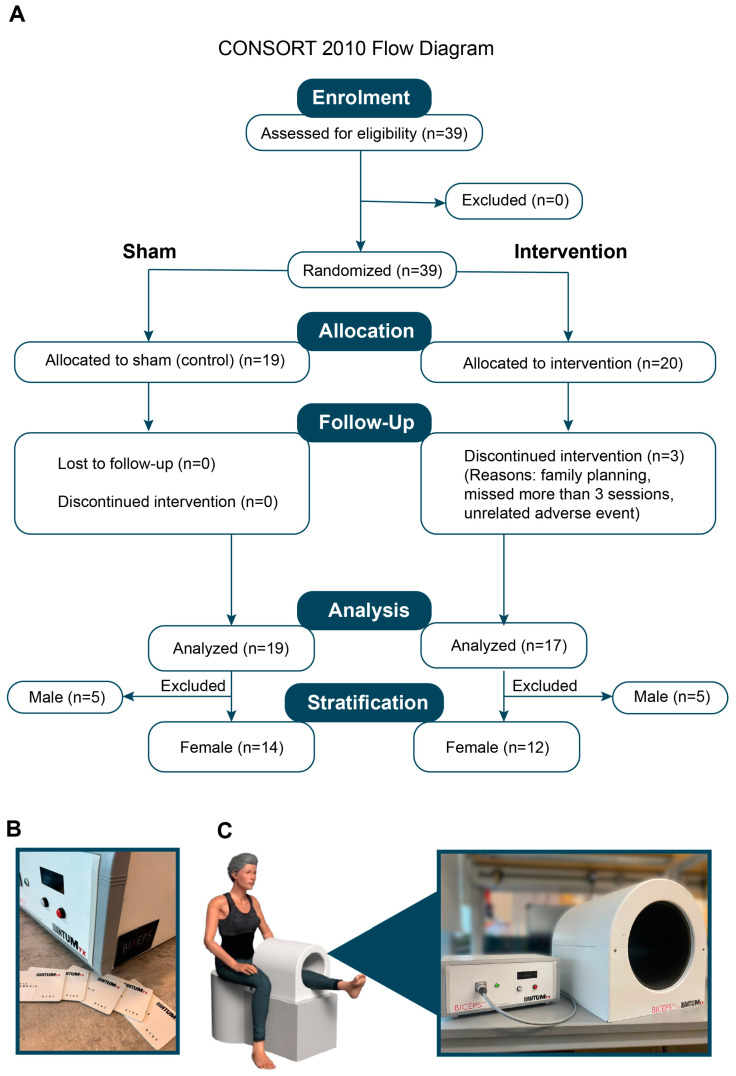
(**A**) The CONSORT flow diagram illustrates the study enrolment, randomization and group allocation. (**B**) The image shows the RFID-activated panel of the controller box accompanied by the corresponding RFID cards. The RFID-controlled system enables double blinding for the trial. (**C**) A graphical illustration showing the position of a subject and their thigh during the intervention. The enlarged image shows the setup of the magnetic device, consisting of a controller box and the magnetic coil.

**Figure 2 cells-14-00331-f002:**
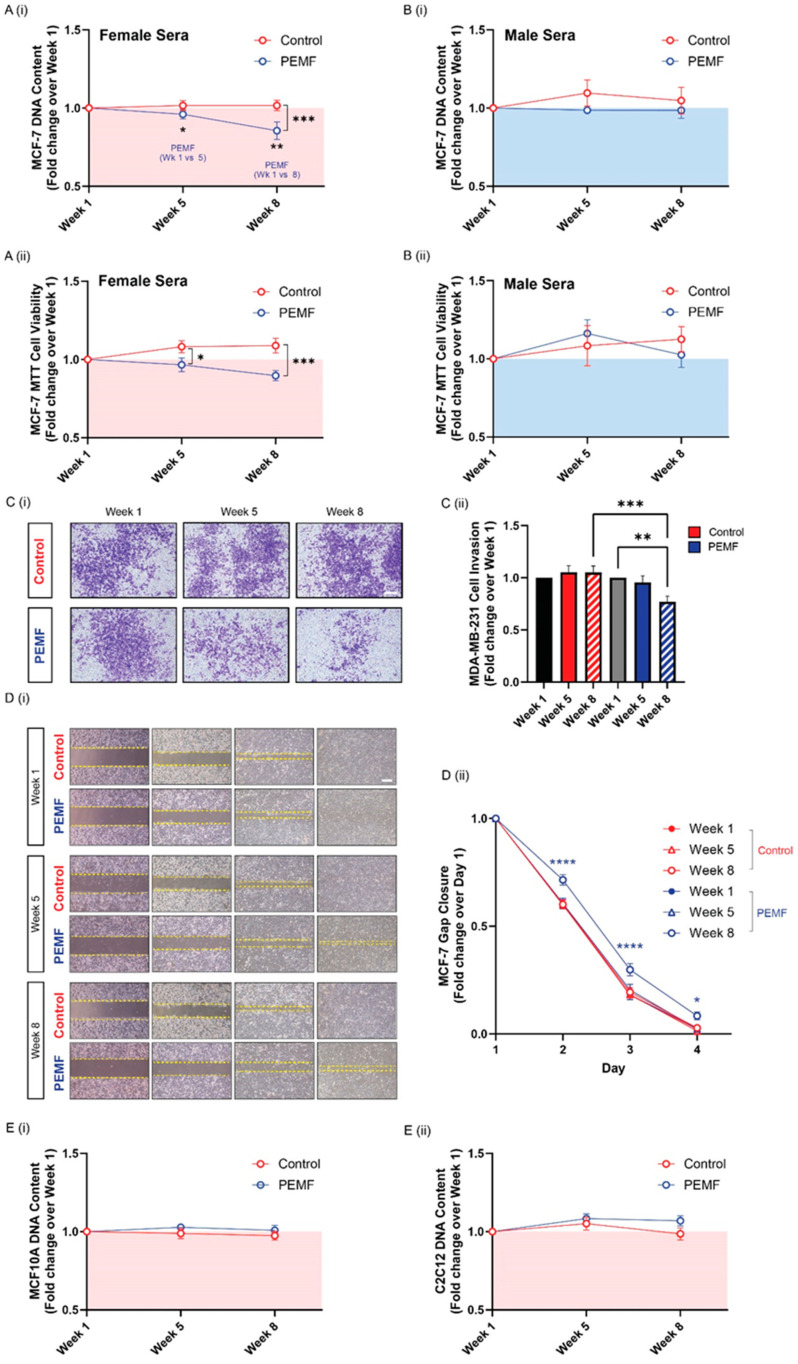
Effect of magnetically exposed human serum on breast cancer cell growth, invasion and migration. (**A**) Changes in the quantification of MCF-7 cell DNA content (**i**) 24 h and (**ii**) MCF-7 MTT cell viability 48 h after the provision of the female sera. Data are expressed as fold change over week 1 of respective sera and the dots represent the average of independent replicates (*n* = 14 control, 12 PEMF). (**B**) Changes in the quantification of MCF-7 cell DNA content (**i**) 24 h and (**ii**) MCF-7 MTT cell viability 48 h after the provision of the male sera. Data are expressed as fold change over week 1 of respective sera and the dots represent the average of independent replicates (*n* = 5 control, 5 PEMF). (**C**) (**i**) Representative images of invading (blue-stained) MDA-MB-231 breast cancer cells 48 h following treatment with female sera of control and PEMF Treatment conditions over the different timepoints of weeks 1, 5 and 8. Scale bar = 300 μm (**ii**) Fold change of invading MDA-MB-231 cells normalized to week 1 of their respective conditions (*n* = 14 control, 12 PEMF). (**D**) (**i**) Brightfield images showing cell gap closure over 4 days following the administration of the indicated human female sera 24 h after the plating of MCF-7 cells for both control and PEMF treatment conditions at separate timepoints of weeks 1, 5 and 8. Scale bar = 300 μm (**ii**) Fold change in MCF-7 cell gap closure in response to the indicated human sera over 4 days (*n* = 14 control, 12 PEMF). (**E**) Changes in the quantification of (**i**) MCF10A and (**ii**) C2C12 cell DNA content 24 h after the provision of the indicated female sera. Data are expressed as fold change over week 1 of respective sera and the dots represent the average of independent replicates (*n* = 14 control, 12 PEMF). Statistical analyses were performed minimally in three independent replicates with * *p* < 0.05, ** *p* < 0.01, *** *p* < 0.001 and **** *p* < 0.0001. The error bars represent the standard error of the mean. Unless otherwise stated, dots represent independent biological replicates.

**Figure 3 cells-14-00331-f003:**
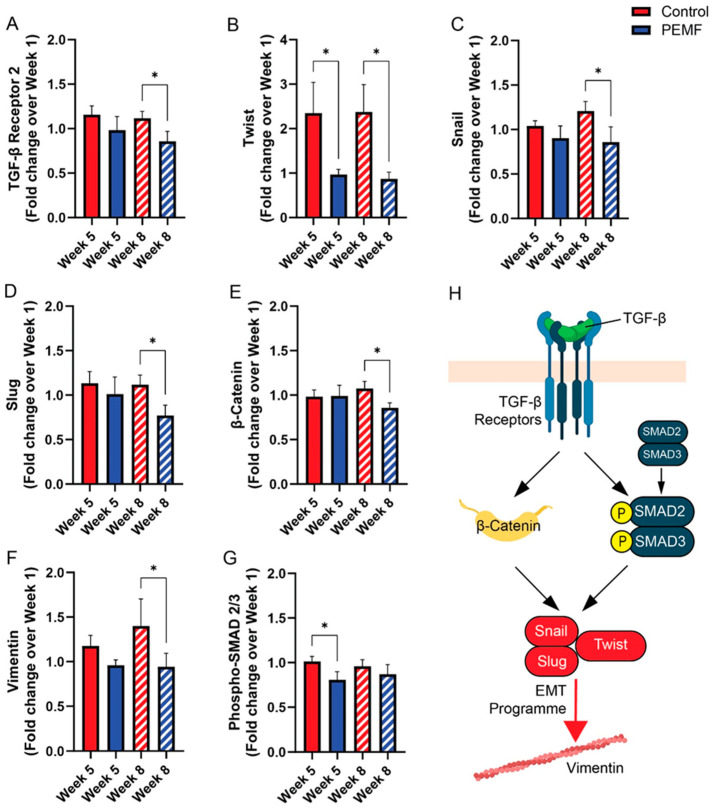
Western Blot Analysis of TGF-β-Associated EMT Markers in MCF-7 Breast Cancer Cells. Western blot analysis of MCF-7 breast cancer cells treated with female human serum from control or magnetic groups for 3 days. Bar charts represent pooled data from cancer cell lysates showing protein expression levels of (**A**) TGF-β receptor 2 (TGFβ2), (**B**) Twist, (**C**) Snail, (**D**) Slug, (**E**) β-catenin, (**F**) Vimentin and (**G**) phosphorylated SMAD 2/3 for week 5 and week 8 comparisons. Data with were normalized and expressed as fold change relative to week 1 values. Statistical analysis was performed using one-way ANOVA with Šidák’s multiple comparisons test (* *p* < 0.05). Error bars represent standard error of the mean. Data represent independent breast cancer cultures (*n* = 12–14 biological replicates per condition per week) exposed to individual human sera. (**H**) Schematic representation of TGF-β signaling pathways in breast cancer cells. TGF-β: Transforming growth factor beta; Snail: Zinc finger protein SNAI1; Slug: Zinc finger protein SNAI2; Twist: Twist-related protein 1.

**Figure 4 cells-14-00331-f004:**
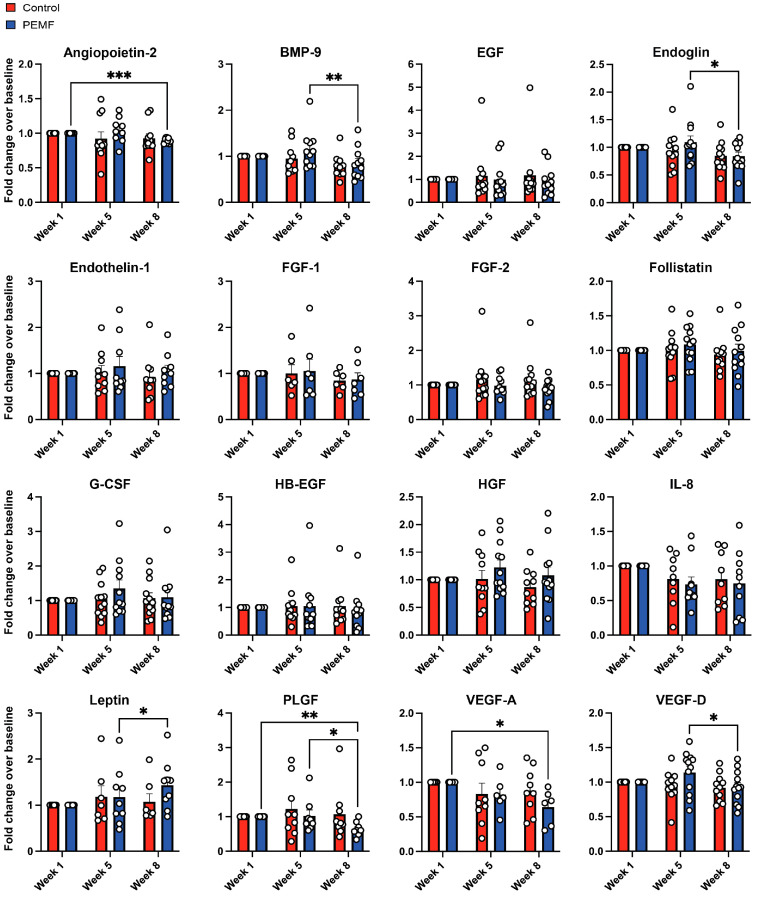
Quantitative analysis of angiogenic factors in response to PEMF therapy using multiplex assay. Changes in angiogenic factors of individual female participants across each timepoint were measured using a human oncology growth factor immunoassay. Data are expressed as fold change over week 1 of respective sera per week and the dots represent independent paired biological replicates. Statistical analyses were performed with * *p* < 0.05, ** *p* < 0.01 and *** *p* < 0.001. Error bars represent the standard error of the mean.

**Figure 5 cells-14-00331-f005:**
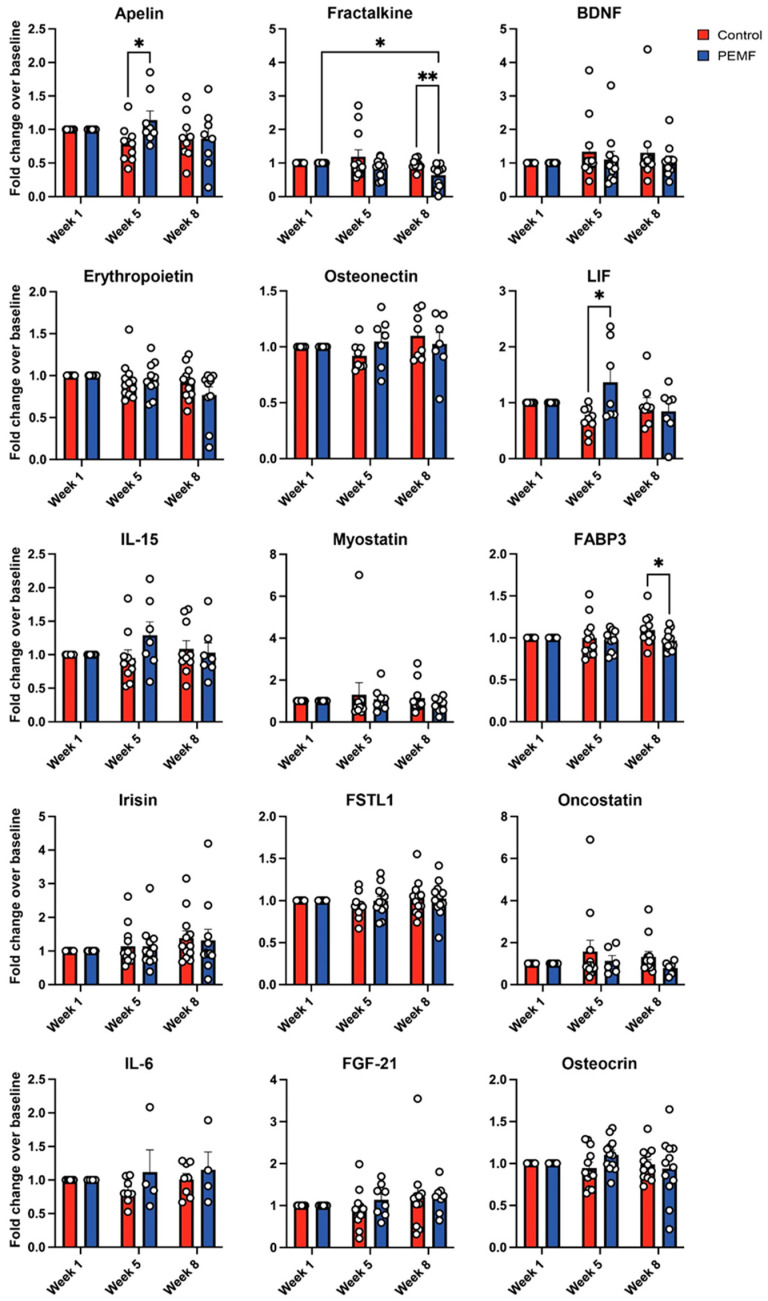
Quantitative analysis of myokines in response to PEMF therapy using multiplex assay. Changes in myokine levels of individual female participants across each timepoint were measured using myokine magnetic bead panel multiplex. Data are expressed as fold change over week 1 of respective sera per week and the dots represent independent paired biological replicates. Comparisons with statistical differences are indicated by * *p* < 0.05 and ** *p* < 0.01. Error bars represent the standard error of the mean.

**Table 1 cells-14-00331-t001:** List of primary antibodies used for Western blot analysis.

Antibody Name	Dilution Factor	Cat. No.	Manufacturer
Vimentin	1:300	sc-373717	Santa Cruz
Twist	1:300	sc-81417	Santa Cruz
Phospho-Smad2/Smad3	1:1000	8828S	Cell Signaling
Smad2/3	1:1000	3102S	Cell Signaling
TGF-β Receptor II	1:1000	66636-1-lg	Proteintech
β-Catenin	1:1000	17565-1-AP	Proteintech
SNAI1	1:1000	13099-1-AP	Proteintech
SNAI2	1:1000	12129-1-AP	Proteintech
GAPDH	1:10,000	60004-1-1g	Proteintech
α-tubulin	1:10,000	66031-1-1g	Proteintech
β-actin	1:10,000	60008-1-1g	Proteintech

**Table 2 cells-14-00331-t002:** List of analytes used for multiplex assay.

Kit	Analyte	Abbreviation	Cat. No.
Human Myokine Magnetic Bead Panel (HMYOMAG-56K) Merck Millipore, Burlington, MA, USA	Apelin	APLN	HAPLN-MAG
	Fractalkine	CX3CL1	HMYFKN-MAG
	Brain-Derived Neurotrophic Factor	BDNF	RBDNF-MAG
	Erythropoeitin	EPO	HEP0-MAG
	Osteonectin	SPARC	H0STNCTN-MAG
	Leukaemia Inhibitory Factor	LIF	HMYLIF-MAG
	Interleukin 15	IL-15	HMYIL15-MAG
	Myostatin/GDF8	MSTN	HMYSTN-MAG
	Fatty Acid Binding Protein 3	FABP3	HFABP3-MAG
	Irisin	IRISN	HIRISN-MAG
	Follistatin-Like 1 Protein	FSTL-1	HFSTL1-MAG
	Oncostatin M	OSM	H0SM-MAG
	Interleukin 6	IL-6	HMYIL6-MAG
	Fibroblast Growth Factor 21	FGF21	HFGF21-MAG
	Osteocrin/Musclin	OSTN	H0STCRN-MAG
Human Angiogenesis/Growth Factor Magnetic Bead Panel 1 (HAGP1MAG-12K) Merck Millipore, Burlington, MA, USA	Angiopoietin-2	ANG-2	HANGPT2-MAG
	Bone Morphogenetic Protein-9	BMP-9	HBMP9-MAG
	Epidermal Growth Factor	EGF	HAGEGF-MAG
	Endoglin	ENG	HENDGLN-MAG
	Endothelin-1	ET-1	HET1-MAG
	Fibroblast Growth Factor 1	FGF-1	HFGF1-MAG
	Fibroblast Growth Factor 2	FGF-2	HFGF2-MAG
	Follistatin	FST	HFLSTN-MAG
	Granulocyte Colony Stimulating Factor	G-CSF	HAGGCSF-MAG
	Heparin-binding EGF-like Growth Factor	HB-EGF	HHBEGF-MAG
	Hepatocyte Growth Factor	HGF	HHGF-MAG
	Interleukin 8	IL-8	HIL8-MAG
	Leptin	LEP	HCCLPTN-MAG
	Placental Growth Factor	PLGF	HPLGF-MAG
	Vascular Endothelial Growth Factor-A	VEGF-A	HVEGF-MAG
	Vascular Endothelial Growth Factor-D	VEGF-D	HVEGFD-MAG

**Table 3 cells-14-00331-t003:** Baseline demographics, anthropometrics, and lifestyle factors of the study cohorts.

	Control (Sham)	PEMF
Sample Size	19	17
Median age (SD)	34 (4)	36 (5)
Gender (Male/Female)	5/14	5/12
Median weight (kg) (SD)	55.5 (15.7)	61 (20.4)
Median height (m) (SD)	1.60 (0.08)	1.63 (0.09)
Median body mass index (SD)	22.3 (4.6)	23.2 (6.3)
Median steps per day	5000–9999	5000–9999
Median sitting hours per day (SD)	8 (1.7)	6 (1.9)

## Data Availability

All data supporting the results are presented in the manuscript. Any other inquiries can be directed to the corresponding authors via email.

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
