# Peer review of "One Month of Brief Weekly Magnetic Field Therapy Enhances the Anticancer Potential of Female Human Sera: Randomized Double-Blind Pilot Study"

_cells, 2025, doi:10.3390/cells14050331_

Round 1

Reviewer 1 Report

Comments and Suggestions for Authors

It is inconceivable that the authors cannot find a few lines to specify in some detail the nature of the 1 mT pulse trains used in their experiments.

I am skeptical that the exposure system using a succession of magnetic pulses peaking at 1 mT would not produce some detectable noise (140).

Most EMFs can interact with mitochondrial respiration by a modification of the distance between OXPHOS complexes, likely inducing a more oxidative phenotype ultimately manifested in blood secretions.

A valuable contribution of the article is to solidify the action of EMR on serum action 1 month after exposure.

I am very surprised that the field "appears to preferentially possess anti-breast cancer activity" while ignoring cancer cells: could the intervention really be so specific?

But then, the clear sex difference also raises eyebrows.

I am very skeptical of the reserves expressed on the practicality (475) of field application to the body. If a therapy can be designed, the practical aspects of field application in the clinic or at home would be the easy part.

This is a well though-out and valuable article, but disturbing biological systems using EMR has always been relatively easy.

What is more difficult is to optimize the field, exposure and biological parameters to yield clinically useful methods. I am sure the authors realize they are tackling a difficult and complex problem, even if they chose to deal exclusively with TGF-β signaling.

For the future, the authors could think of controlling the oxygen levels in their diagnostic cultures, as it may impact the size of the effects observed.

Another difficulty is that even in the presence of a viable therapeutic method, PEMFs are very likely to have substantial side effects outside the cancer cell suppression circle.

Reviewer 2 Report

Comments and Suggestions for Authors

This study demonstrated that the blood from 20 healthy female subjects having undergone one month of biweekly muscle-targeted PEMF therapy presented with anticancer attributes when tested in vitro, in the human sera on cancer cell dynamics, cancer signaling change, TGFb associated EMT markers, angiogenic factors, and myokines. It will be better to demonstrate any biochemical or cellular change or similarity in the treated group.   
